# Effect of Rice Bran and Retrograded Time on the Qualities of Brown Rice Noodles: Edible Quality, Microstructure, and Moisture Migration

**DOI:** 10.3390/foods12244509

**Published:** 2023-12-17

**Authors:** Hong Feng, Ting Li, You Zhou, Qingyun Lyu, Lei Chen, Xuedong Wang, Wenping Ding

**Affiliations:** 1School of Food Science and Engineering, Wuhan Polytechnic University, Wuhan 430023, China; sinlio0909@163.com (H.F.); 17616286182@163.com (T.L.); m17786792286@163.com (Y.Z.); chenleiy@whpu.edu.cn (L.C.); xuedongwuhan@163.com (X.W.); whdingwp@163.com (W.D.); 2Key Laboratory of Grain and Oil Processing, Ministry of Education, Wuhan 430023, China

**Keywords:** brown rice noodle, rice bran, retrogradation, edible quality, moisture migration

## Abstract

Brown rice, as a kind of whole-grain food, has attracted significant attention due to its health benefits. This paper aimed to investigate the effect of rice bran content and retrograded time on the physicochemical properties and culinary qualities of brown rice noodles (BRNs). The results indicated that the addition of rice bran altered the pasting properties, gel properties, and texture of the brown rice flours (BRFs). The optimal cooking time and water absorption of BRNs were reduced after the incorporation of rice bran to 14.9% and 41.9%, respectively, while the breaking rate increased from 2.2% to 23.3%. The color of BRNs became darker and yellower, and the overall acceptability by the consumer decreased. The addition of rice bran also led to a decrease in hardness, chewiness and crystallinity. The binding water inside the BRNs decreased, while the free water increased, resulting in a looser structure. This study revealed that the retrograded time of the BRNs also affected its quality. When the retrograded time was 7 h, the cooked BRNs had a lower breaking rate, good hardness, cohesiveness, chewiness, and better overall acceptability by consumers. The structure was compact, the internal binding water content of BRN was higher, and the free water content was lower. This study provides insights into developing nutritionally healthy, high-quality novel rice flour products, and offers a theoretical basis for the industrial production of BRNs.

## 1. Introduction

Brown rice, known for its rich nutritional content and numerous health benefits, has indeed gotten attention in recent years. Its consumption has been linked to better metabolic function, gluten-free properties, hypoallergenic qualities, low fat content, easy digestibility, and risk reduction effects of various health conditions such as high blood sugar, cancer, obesity, and cardiovascular disease [1,2,3]. Rice noodles, as the most produced and consumed rice product, are particularly popular in Asian countries [4]. This further emphasizes the potential of brown rice flour to become an essential ingredient in whole-grain foods, given its numerous health benefits and widespread consumption. As the demand for functional and healthy foods continues to grow, brown rice flour has the potential to play a significant role in meeting these needs. Its versatility in various food products, such as noodles, bread, and other baked goods, makes it a promising ingredient for the food industry to explore and incorporate into their product offerings. Therefore, brown rice flour (BRF) has great potential to play an important role in the functional and health food market due to its numerous health benefits and widespread consumption, which should be considered to meet the growing demand for healthier food options by consumers.

Rice bran is an important byproduct of the rice milling industry, with a global potential of 29.3 million tons annually. It has gained great attention of the researchers due to its nutrient-rich composition, easy availability, low cost, high antioxidant potential, and promising effects against several metabolic ailments [5]. Rice bran contains dietary fiber, γ-glutamic acid and γ-aminobutyric acid, which has been reported to be effective in preventing hypertension and type II diabetic disease, making it a promising low-glycemic index (GI) food option [6]. However, the addition of rice bran has an impact on the quality of food. Investigations have demonstrated that the incorporation of rice bran into bread resulted in alterations to the texture of bread and nutritional profile. However, at an addition level higher than 10% of rice bran (both full-fat and defatted rice bran) or with crude rice bran fiber, it negatively affected the specific volume and texture characteristics [7]. The method of extrusion cooking could improve the storage quality of fresh BRF prepared from rice bran and inhibit starch retrogradation during storage, which was beneficial to improve storage quality [8]. Adding expanded rice bran to rice noodles aiming to improve the cooking and edible quality of rice noodles, resulted in the lowest starch digestibility and the highest content of resistant starch [9]. Retrogradation is one of the important reasons that affects the quality of rice products, and the degree of retrogradation significantly affects the quality of rice noodles [10]. Retrogradation can effectively reduce the cooking loss and improve the textural properties of rice noodles [11]. However, studies on the quality of brown rice noodles (BRNs) with different rice bran contents and different retrograded times are rare.

The purpose of this study was to investigate the effects of different rice bran contents on the properties of BRNs made from a mixture of white rice flour and rice bran, and retrograded time on the quality of BRNs. In this study, rice bran was separated from brown rice and treated separately, and then the rice bran was added into white rice flour to analyze the effect of rice bran contents on the physicochemical properties of BRFs. They were made into BRFs with different retrograded times of whole brown rice flours and used in the study to investigate the quality effects. Properties of BRNs such as cooking properties and texture were studied, while the microstructure and water migration of BRNs were investigated. The results of this study could provide a theoretical basis for incorporating BRNs into commercial production to promote the consumption of brown rice versus white rice.

## 2. Materials and Methods

### 2.1. Materials

Brown rice (Zhenzhu rice, China) was provided by Xidi (Wuhan city, China) Agriculture Co., Ltd. The wet basis was 12.9 ± 0.2%. All reagents used in this study were of analytical grade.

### 2.2. Preparation of BRFs

The brown rice was milled into fine white rice and rice bran (with a milled rice rate of 11%) using a rice milling machine (TM05C-C, Satake Machinery (Suzhou) Co., Ltd., Suzhou, China). Rice bran was crushed using a hammer cyclone mill (JXFM110, Shanghai Jiading Grain and Oil Apparatus Co., Ltd., Shanghai, China), and passed through an 80-mesh sieve to obtain rice bran flour. Fine white rice was crushed using a universal grinder (30B, Changzhou Dingzhuo Drying Equipment Co., Ltd., Changzhou, China) to obtain white rice flour with a particle size less than 150 μm. The white rice flour was mixed with rice bran flour for measuring the quality of mixed flour and processed into BRFs at ratios of 100:0, 97:3, 94:6, 91:9, and 89:11 to form BRNs, named BRF100, BRF97, BRF94, BRF91, and BRF89, respectively.

### 2.3. Preparation of BRNs

Using the twin-screw extruder (Jiangxi Huadachang Food Co., Ltd., Ganzhou, China), the variant of BRFs was transformed into bran rice noodle BRNs, and the moisture content was adjusted to 38%. The fresh BRNs were placed into the artificial climate chamber (RH-LHP-300L, Changzhou Runhua Electric Appliance Co., Ltd., Changzhou, China). BRF100, BRF97, BRF94, BRF91, and BRF89 were made into BRNs, named BRN100, BRN97, BRN94, BRN91, and BRN89, accordingly. Retrograded BRN89 at was treated at 4 °C and 75–90% humidity for 1, 3, 5, 7, and 9 h, respectively, labeled as BRN1h, BRN3h, BRN5h, BRN7h, and BRN9h. Subsequently, the BRNs were dried to ensure a moisture content of 13% or lower.

### 2.4. Physical and Chemical Indicators of BRFs

#### 2.4.1. Solubility and Swelling Power

The solubility (*S*) and swelling power (*SP*) of BRFs were measured according to Fu et al. [12]. Of the raw material, 800 mg (*W*_0_) was mixed well with 25 mL of distilled water in a 50 mL centrifuge tube and equilibrated in a water bath at 25 °C for 5 min. The equilibrated samples were shaken and kept at a constant temperature in a water bath at 95 °C for 30 min, followed by an ice bath for 1 min, then left to equilibrate at 25 °C for 5 min and centrifuged using a centrifuge (TDZ5-WS, Changsha Pingfan Instrument Co., Ltd., Changsha, China) at 4000 rpm for 20 min. The supernatant was transferred to an aluminum box, dried in an oven at 105 °C to a constant weight (*W*_1_), and the precipitate was weighed (*W*_2_). The *SP* and *S* of the samples were calculated according to the following Equations (1) and (2):(1)$$S=\frac{W1}{W0}\times 100$$
(2)$$SP=\frac{W2}{W0(100-S)}\times 100$$

#### 2.4.2. Pasting Properties

A Rapid Visco Analyzer (RVA Super 4, Newport Scientific Pty Ltd., Warriewood, Australia) was used to measure the pasting properties of the BRFs based on the method of AACC 61-02 (AACC, 2002).

#### 2.4.3. Properties of BRF Gel

The gel properties of the BRFs were determined using the method referring to by Jia et al. [13]. The different BRFs were suspended in water at a 10% (*w*/*w*) concentration and placed in a boiling water bath shaking incubator (ZHSY-50, Shanghai Zhi Chu Instruments Co., Ltd., Shanghai, China) for 20 min, then removed and placed in a refrigerator at 4 °C for 24 h. The BRF gel was analyzed qualitatively using a physical property tester (TA-XT2i, Stable Micro System, Ltd., Godalming, UK), probe P 0.5R, and tested twice for compression at 50% deformation with a pre-test speed of 2.0 mm/s, mid-test speed of 1.0 mm/s, and post-test speed of 2.0 mm/s. Each sample was tested three times, and the average value was taken. The hardness, adhesiveness, cohesiveness, springiness, gumminess and chewiness of these samples were recorded.

### 2.5. Edible Quality and Microstructure of BRNs

#### 2.5.1. Cooking Quality

Twenty BRNs of similar quality (uniform thickness and length of BRNs) were selected and placed in a beaker with 500 mL of boiling water and timed with a stopwatch. After cooking for 6 min at a slight boiling state, one sample was taken out every 15 s and placed in two colorless transparent glass pieces and squeezed to observe whether the white core in the center of the noodle disappeared. When the white core completely disappeared, the cooking was stopped and the optimal cooking time was recorded [14]. Thirty BRNs with uniform quality and a length of 20 cm were boiled in boiling water until the optimal cooking time was reached. Then, all BRNs were taken out and the quantity (*N*) recorded was used to calculated the breaking rate (*D*); the calculation method is shown in Formula (3). Twenty BRNs were taken, weighed (*m*_1_), put into 1000 mL of boiling water, and steamed in a slightly boiling state, cooked for the optimal cooking time, then taken out and rinsed slowly with running water for 30 s. After absorbing the excess moisture on the surface of the BRNs, they were weighed and this was expressed as *m*_2_ and the initial moisture content of BRNs as *W*. The measurement was repeated three times, and the average value was taken. The water absorption rate (*L*) of the BRNs was calculated according to Equation (4) [9].
(3)$$D\%=\frac{N-30}{30}\times 100$$
(4)$$L\%=\frac{m2-m1(1-W)}{m1(1-W)}\times 100$$

#### 2.5.2. Texture Properties of BRNs

The texture properties of BRNs during cooking were analyzed using TA-XT 2i/5 Texture Analyzer (Stable Micro System, Ltd., Godalming, UK) following a reported method of Geng et al. [15] with several modifications. The full texture testing was performed on the sample using Texture Profile Analysis (TPA) mode and the P/45 probe selected. Three high-quality BRNs were used for one test with a pre-test speed of 2.0 mm/s, mid-test speed of 1.0 mm/s, and post-test speed of 2.0 mm/s. The displacement of this test was 75 mm, with a strain of 50% and a trigger force of 5.0 g. Each set of samples was tested five times, and the average value was taken. The hardness, cohesiveness, and chewiness of the rice noodles were determined.

#### 2.5.3. Chromaticity Characteristics

Of the dried BRNs, 10 g was crushed using a universal grinder, with a grinding time of 60 s. The powdered BRNs were then measured for color using a portable colorimeter (cs-10, manufactured by Liangchuang Instruments (Suzhou) Co., Ltd., Suzhou, China). The color parameters include *L* for lightness (0–100), *a* for the red–green value (positive values represent red and negative values represent green), and *b* for the yellow–blue value (positive values represent yellow and negative values represent blue). *W*, for whiteness, was calculated as follows:(5)$$W=100-{\left({\left(100-L\right)}^{2}+a^{2}+b^{2}\right)}^{\frac{1}{2}}$$

#### 2.5.4. Sensory Evaluation

The sensory evaluation of the BRNs was carried out by referring to Yadav’s method [16]. Of the BRNs, 200 g was added to 2 L of boiling water, cooked to the optimal cooking time, and the BRNs strained for sensory evaluation. The sensory panel comprised 6 trained members who were selected from the university community and aged 21–25 years (3 males and 3 females). The panelists were trained with a hedonic scale and texture parameters for sensory analysis. All the noodle samples were coded with three random numbers before being presented to the panel. Purified drinking water was used to clean the mouth in between the samples. The firmness, chewiness, elasticity, slipperiness, and overall acceptability of BRNs were assessed (in Table 1). All volunteers were informed about the content of the experiment. All volunteers volunteered to participate in this sensory evaluation and agreed to have the collected data counted. The rights and privacy of all participants were protected during the execution of the study. This experiment did not require approval from the Ethics Committee because there were no risks associated for panelists who tasted samples, and this experiment met the national standards of the People’s Republic of China.

#### 2.5.5. X-ray Diffraction (XRD)

The BRNs were ground and sieved, and their moisture was equilibrated for 24 h. The crystallinity of the starch in the BRNs was determined using an X-ray diffractometer (Empyrean, Ruijin, The Netherlands) method referring to by Chi et al. [17].

#### 2.5.6. Low-Field Nuclear Magnetic Resonance (LF-NMR)

The samples were measured using a nuclear magnetic resonance imaging analyzer (NMI20-040V-I, Suzhou Newmark Analytical Instruments Co., Ltd., Suzhou, China). Samples of uniform quality were selected and put into special glass tubes for testing. After instrument calibration, the glass tube was inserted into the test tank, and the test was performed via computer software operation. The test parameters were as follows: probe selection 40 mm, SW: 200 KHz, RFD: 0.002 ms, temperature: 32 °C, field strength: 20 MHz, sampling method: cumulative sampling. The spin relaxation time, T2, of the meter line was measured and inverted using the CPMG [18] sequence method.

#### 2.5.7. Microstructures of the BRNs

The microstructure of the BRNs was observed via the method referring to by Geng et al. [15], with several modifications. The BRNs were cut into 1 mm slices, placed individually on a sample holder, and coated with gold. All samples were observed and photographed using a field emission scanning electron microscope (ZEISS Sigma300, Zeiss AG, Oberkochen, Germany) at an accelerating voltage of 10.0 kV. The transverse relaxation time, T_2_, was determined using the CPMG (Carr–Purcell–Meiboom–Gill) pulse sequence.

### 2.6. Statistical Analysis

SPSS software was used for one-way analysis of variance (ANOVA) with Duncan’s multiple-range test (*p* < 0.05). All experiments in this study were performed at least in triplicate. Origin Pro version 9.0 was used for image processing.

## 3. Results and Discussion

### 3.1. Physical and Chemical Indicators of BRFs

#### 3.1.1. Solubility and Swelling Power of BRFs

From Figure 1. Solubility and swelling power of BRFs. Pairs of samples labeled with different letters in the same index indicate significant differences (*p* < 0.05), it could be seen that as the proportion of rice bran increased, the *S* of BRFs significantly increased, and the *SP* of BRFs significantly decreased. After crushing rice bran, the dietary fiber particles were micronized, making it easier to embed them into the interior of starch molecules and destroy their original structure, which might be the reason for the increase in solubility [19,20,21]. Previous studies have shown that the *SP* depends on the size of the interaction between the crystalline domain and the starch chains in the amorphous domain [22]. As the proportion of rice bran increased, the relative content of starch decreased, and therefore, the *SP* decreased.

#### 3.1.2. Pasting Properties of BRFs

The pasting properties of BRFs were regarded as crucial parameters to evaluate the cooking quality of BRNs [15]. As shown in Figure 2 and Table 2, with the increase of rice bran content, the peak viscosity, trough viscosity, and breakdown value of BRFs all showed a decreasing trend, while the setback value and pasting temperature showed an overall increasing trend. There was no significant difference in peak time. The reason might be the increase in rice bran content. The dietary fiber and lipids in rice bran inhibited the combination of starch molecules and water during the starch gelatinization heating process, thereby suppressing starch gelatinization, leading to a decrease in peak viscosity and an increase in pasting temperature. These results were similar to the research results of Chung et al. [23]. Dietary fiber in rice bran restricted the expansion of starch, reduced the pasting viscosity of BRFs, and led to high cooking losses and a poor texture of BRNs.

#### 3.1.3. Gelation Properties of BRFs

The gelation properties of BRFs could reflect the quality of BRNs to a certain extent, and they were highly correlated. As shown in Table 3, the hardness, adhesiveness, springiness, and chewiness of the BRF gel decreased significantly with the increase in the proportion of rice bran. The hardness, adhesiveness, springiness, and chewiness of BRF89 decreased by 28.9%, 54.5%, 30.3%, and 25.5% (*p* < 0.05) compared with BRF100, respectively. There were no significant differences in cohesiveness and gumminess. This phenomenon could be attributed to the presence of substantial molecules, such as proteins and amino acids in rice bran. These molecules interfered with the binding of starch molecules with water following the addition of rice bran. Consequently, the partial starch pasting was inadequate, which impeded the regeneration and rearrangement of the gel structure [24,25,26].

### 3.2. The Edible Quality of BRNs

#### 3.2.1. Cooking Quality

The water absorption and breaking rate of BRNs significantly affected its edible quality. As shown in Table 4, with the content of rice bran increased, the optimal cooking time and water absorption rate of BRN89 were significantly reduced by 14.9% and 41.9% (*p* < 0.05) compared with BRN100, respectively. The breaking rate of BRN89 was elevated by 9.51% compared with BRN100. These results indicated that the cooking quality of BRNs gradually decreased with the addition of rice bran. During the cooking process, the dietary fiber in rice bran promoted the dissolution of starch and other substances in BRNs, and blocked the formation of a starch gel network. This led to an increase in breaking rate and the BRNs were more prone to paste soup, which was consistent with previous studies [27,28].

As shown in Table 4, the optimal cooking time of BRNs showed an increasing trend with the increase in retrograded time, from 6.42 min (BRN1h) to 9.97 min (BRN9h) (*p* < 0.05). The water absorption and breaking rate decreased with increasing retrograded time. The structure of BRNs gradually became tighter during retrogradation with the increase in time, and water molecules were not easy to enter the interior of BRNs during cooking. It was also consistent with the fact that the water absorption and breaking decreased with the time increasing, which showed that the BRNs were more resistant to cooking with the extension of retrograded time.

#### 3.2.2. Textural Properties of BRNs

As shown in Table 5, the hardness, cohesiveness, and chewiness of the BRNs decreased significantly as the proportion of rice bran increased. Hardness is usually considered an indicator of the overall quality of rice noodles. The hardness decreased by approximately 57% (*p* < 0.05), falling from 3001.44 g (BRN100) to 1300.62 g (BRN89). Cohesiveness, defined as the degree to which food clumps together after chewing, decreased by 5.7% for BRNs. Based on the decline in all indexes, it could be inferred that an increase in rice bran proportion inhibits starch molecule cross-linking, disrupted the spatial configuration of BRNs, and markedly diminishes their quality. It has been reported that excess dietary fiber may deteriorate the interactions between starch molecules [29]. This was consistent with the results of cooking quality.

As shown in Table 5, with the increase in retrograded time, the hardness increased by approximately 28%, from 1022.05 g (BRN1h) to 1311.40 g (BRN9h), and there was no significant difference in cohesiveness (*p* < 0.05). On the other hand, the chewiness saw a significant increase of 31.7%, rising from 626.05 (BRN1h) to 824.64 (BRN9h). Under the suitable temperature and humidity conditions, the starch molecules in brown rice flour retrograded and arranged in an orderly manner with the extension of retrograded time, forming a better gel network structure. The hardness gradually increased, and the chewiness increased to some extent.

#### 3.2.3. Chromaticity Characteristics of BRNs

The color and visual appeal of rice noodles are crucial factors influencing consumer acceptance. As depicted in Table 6, the L, a, and W values of BRNs progressively decreased, while the b value increased as rice bran content increased. This indicated that the color of BRNs turned darker and more yellowish. Consequently, adding rice bran had a detrimental effect on the color of BRNs. The reason for this is that rice bran naturally contains various pigment substances, and the higher the bran content, the deeper the resulting BRN color. This change can be attributed to the natural pigments present in brown rice, such as polyphenols and carotenoids [30].

As observed in Table 6, the value W of BRNs progressively declined with increasing retrograded time. The W value dropped from 82.36 (BRN1h) to 77.49 (BRN9h) post-retrogradation, representing a decline of approximately 5.8%. The value L decreased from 88.35 (BRN1h) to 82.67 (BRN9h), experiencing a drop of about 6.4%. In contrast, the b value escalated from 12.24 (BRN1h) to 14.18 (BRN9h), with an increase of roughly 15.8%. It can thus be inferred that retrogradation has a detrimental impact on the color of BRNs. As the retrogradation duration prolongs, the color of BRNs becomes yellower and darker.

#### 3.2.4. Sensory Evaluation

As shown in Table 7, BRN100 had the highest overall sensory evaluation, and the overall acceptability of the BRNs decreased significantly with the increase in the proportion of rice bran, with a decreasing trend in hardness, smoothness, elasticity, and chewiness, and BRN89 was the smallest, which indicated that the more the proportion of rice bran added, the lower the sensory evaluation of the BRNs and the less acceptable to consumers. Some studies have shown that the rice bran layer of brown rice negatively affects the sensory and textural quality of rice noodles [31].

From Table 7, it can be observed that the sensory score of BRNs increases first and then decreases with the increase in retrograded time. Among them, the total score of BRN7h was the highest (79.97). This is because as the aging time prolongs, the hardness, elasticity, and chewiness of the BRNs gradually increase, and the scores of hardness, elasticity, and smoothness can also reflect this. The decrease in score of BRN9h may be due to the prolonged retrograded time, which leads to the deterioration of the taste of the BRNs. Therefore, 7 h is the optimal retrograded time for BRNs.

### 3.3. Microstructure of BRNs

As shown in Figure 3, the cross-section of the BRNs gradually became rough and loose as the rice bran content increased. BRN100 showed a tight and smooth cross section, indicating that the starch was completely pasted and BRNs had a dense structure. With the increase in rice bran proportion, it showed rough cross section and uncompacted structure. The voids and pores increased and the distribution was not uniform, which indicated that a perfect gel structure was not formed inside the BRNs. These results were consistent with the trend of shorter optimal cooking time and greater cooking loss in cooking characteristics and less hardness and elasticity in textural characteristics. Rice flour with few and uniformly distributed air bubbles had high water absorption and low cooking loss [32], which was consistent with this study.

As shown in Figure 3, the number of pores on the cross section of BRN1h was the largest, when the BRNs were not retrograded sufficiently. The interaction between starch molecules was weak and the structure loose, the BRNs were formed mainly due to the forced extrusion by external forces. The water molecules were evenly dispersed in BRNs. After drying, smaller pores appeared. The breaking rate of the BRNs was high after cooking with the extension of retrograded time from BRN3h, BRN5h, and BRN7h. It could be seen from the graphs that the pore number of BRNs on the unit surface area gradually decreased, and the pore area gradually increased. The pores of BRN9h were observed to be the largest, a phenomenon attributed to the prolongation of retrograded time. The starch molecules cross-linked with each other, causing the water molecules to be expelled and congregate together. The number of pores formed by the loss of water in the cross section of the BRNs after drying decreased, but the corresponding holes formed became larger. BRN9h retrograded excessively, and the hardness became larger. The pores were too large, which seriously affects the structure of the BRNs, balancing the number of pores and the size of the pores. Overall, the rice noodles exhibit superior quality when the retrogradation time reaches 7 h.

### 3.4. XRD and Crystallinity

Crystallinity is the proportion of the crystalline phase in the starch structure, which is an important index to characterize the degree of crystallization and order in the crystalline region of starch. The crystallinity of different samples is shown in Figure 4.

In Figure 4a, it can be seen that the degree of crystallization of starch in BRNs gradually decreased with the increase in rice bran content. BRN100 showed typical A+V type crystallization peaks, with peaks around 15°, 17°, 18°, and 23° for A type and 7.4°, 12.9°, 19.8°, and 22.4° for V type, respectively. BRN89 showed typical V-type crystallization peaks with sharper peak shapes, which indicated that the increase in rice bran did not change the type of crystal, but the addition of rice bran promoted the binding of starch with other macromolecules and inhibited the regeneration of starch. The crystallinity decreased gradually and significantly with the increase in the percentage of rice bran addition, and the crystallinity of BRN89 decreased by 12.1%. The increase in rice bran content led to the relative decrease in starch content and the chance of mutual cross-linking of starch molecules for regeneration. Another reason might be due to the formation of fibrous protofibrils and straight-chain lipid complexes [33].

As shown in Figure 4b, the BRNs exhibited a typical V-shaped crystalline structure, with peaks at about 7.4°, 12.9°, and 19.8°, respectively. With the increase in retrograded time, the crystallinity gradually increased by 64% from 1 h to 9 h. At the retrograded time of 9 h, the highest loss rate of BRNs after cooking could be attributed to the longer retrograded time, resulting in more retrogradation of branched-chain starch, longer optimal cooking time, increased leaching of straight-chain starch during cooking, and a looser the structure of BRNs that adversely affected the quality of BRNs.

### 3.5. Water Distribution and Migration of BRNs

The distribution and migration behavior of water in BRNs for different rice bran contents and retrograded times are shown in Figure 5. T_21_ (0.19–0.28 ms), T_22_ (11.55–13.60 ms), and T_23_ (144.80–249.36 ms) indicate the distribution of different types of water in BRNs. Among these Ts, T_21_ is considered a sign of strongly bound water, which was present inside the starch granules. T_22_ was considered weakly bound water, which is present outside the starch granules. T_23_ was considered a sign of free water [34].

As shown in Figure 5a, the content of bound water gradually decreased and the content of non-fluidizable water gradually increased with the increase in rice bran content, and free water even appeared in BRN89 and BRN91. This indicated that the addition of rice bran leads to the weakening of the binding ability of starch molecules and water molecules in BRNs. The increase in free water content in BRNs might lead to the growth of microorganisms, which was unfavorable for the storage and preservation of the products [8].

From Figure 5b, it can be seen that the bound water content gradually increased and the free water content gradually decreased as the retrograded time increased. Under the effect of retrogradation, the macromolecules precipitated due to the change in molecular structure, and the protein in rice bran had a strong ability to bind water, which limited the migration of water, and the structure of BRNs became tight [35]. However, too-long retrograded time led to the hardening of BRNs due to a loss of water and collapse of structure, which was consistent with the results of edible quality, texture, and microstructure.

## 4. Conclusions

In this study, rice bran content had a significant effect on the physicochemical properties of rice flour. The addition of rice bran significantly decreased the pasting viscosity of starch, which affected the formation of starch gel. The hardness, elasticity, and adhesiveness of starch gel decreased significantly. As the proportion of rice bran increased, the optimal cooking time and water absorption of brown rice noodles decreased. Moreover, an increase in breaking rate, hardness, chewiness, crystallinity, and the content of bound water within the molecules or free water was observed. An increase in rice bran content made the structure of brown rice noodles rough and loose, and the overall acceptability by consumers and sensory scores decreased. When the retrograded time was less than 7 h, it led to incomplete retrogradation and significant cooking losses. The cooked texture became harder and less viscoelastic due to excessive retrogradation (more than 7 h). The comprehensive results show that when the retrograded time is 7 h, the optimal cooking time was longer, with better structure, less breaking rate, better hardness, cohesiveness and chewiness, and the highest sensory score. Microstructure showed that the holes in the BRN7h were smaller and less in number at this time, and low-field nuclear magnetic resonance analysis showed that the starch-bound water content was higher and free water content lower. The results and discussion of this study could provide a reference for the production process and quality of brown rice noodles.

## Figures and Tables

**Figure 1 foods-12-04509-f001:**
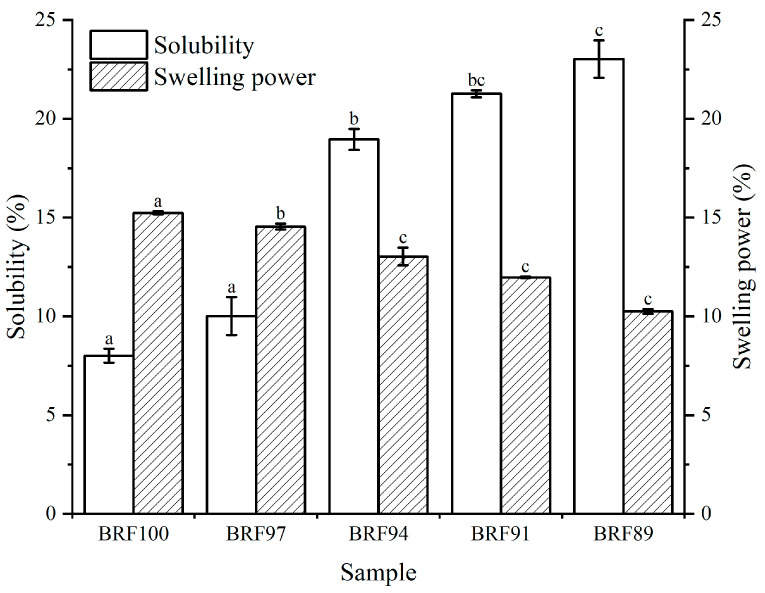
Solubility and swelling power of BRFs. Pairs of samples labeled with different letters in the same index indicate significant differences (*p* < 0.05).

**Figure 2 foods-12-04509-f002:**
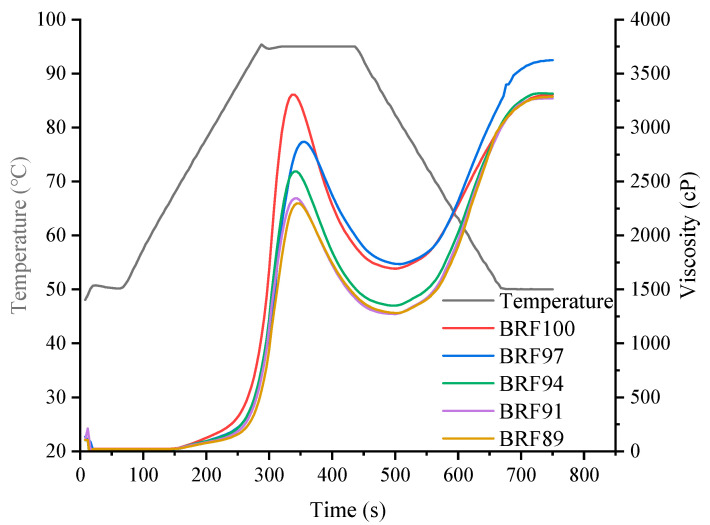
Pasting properties of BRFs.

**Figure 3 foods-12-04509-f003:**
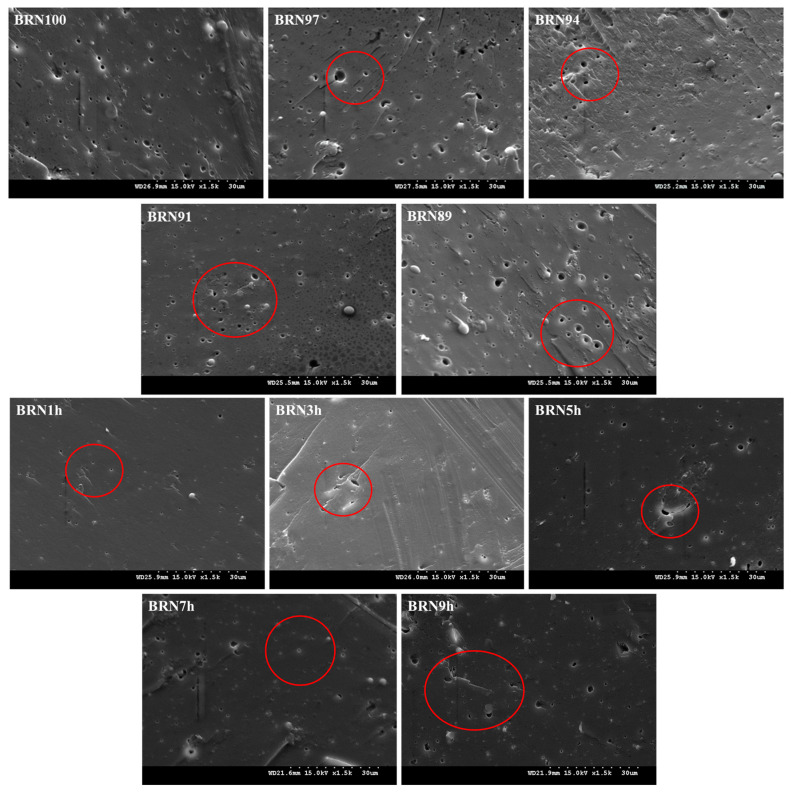
The surface microstructure of BRNs, 1500×.

**Figure 4 foods-12-04509-f004:**
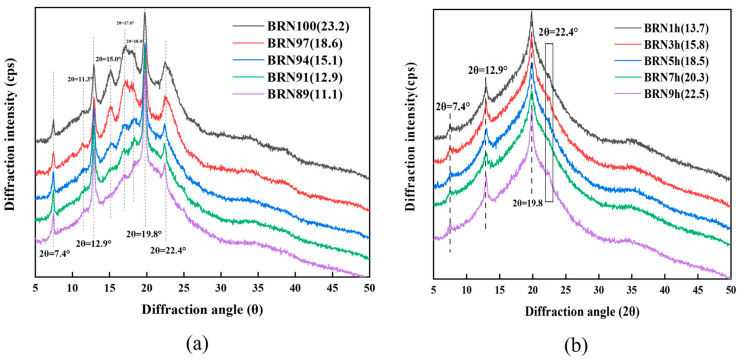
XRD patterns of BRNs, relative crystallinity of BRNs in parentheses and expressed as %. (**a**) Different contents of rice bran in BRNs. (**b**) Different retrograded times of BRNs.

**Figure 5 foods-12-04509-f005:**
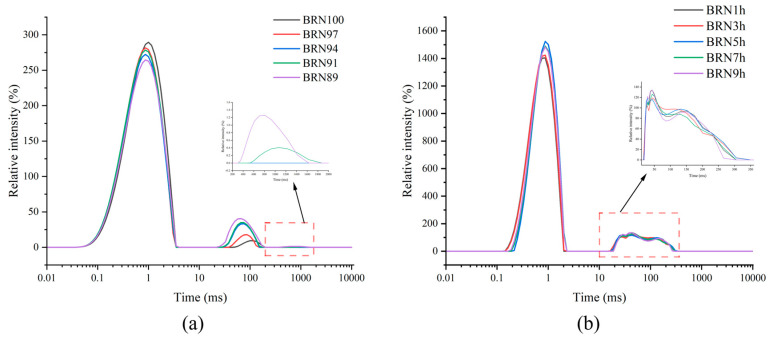
Moisture distribution of BRNs, the subfigure (**a**) showed the changes over times (200–2000 ms) and the subfigure (**b**) showed the changes over times (10–360 ms).

**Table 1 foods-12-04509-t001:** Sensory evaluation of BRFs.

Sensory Attribute	Definition	Score
Firmness	The required force to bite through brown rice noodles.	1~20
Chewiness	Time required for chewing the sample at a constant speed to achieve a swallowing consistency.	1~20
Elasticity	The elasticity felt during tooth occlusion.	1~20
Slipperiness	The feeling of brown rice noodles sliding over your tongue.	1~20
Overall acceptability	Acceptability of the appearance of cooked brown rice noodles.	1~20

**Table 2 foods-12-04509-t002:** Pasting parameters of BRF samples.

Sample	BRF100	BRF97	BRF94	BRF91	BRF89
Peak viscosity/cp	3200.50 ± 59.50 ^a^	2758.50 ± 80.50 ^b^	2569.00 ± 25.00 ^c^	2351.50 ± 6.50 ^d^	2287.50 ± 10.50 ^d^
Trough viscosity/cp	1658.50 ± 3.50 ^a^	1460.00 ± 41.00 ^b^	1346.00 ± 3.00 ^c^	1278.50 ± 6.50 ^cd^	1272.50 ± 8.50 ^d^
Breakdown value/cp	1542.00 ± 63.00 ^a^	1298.50 ± 39.50 ^b^	1240.50 ± 4.50 ^b^	1068.50 ± 4.50 ^c^	1009.50 ± 7.50 ^c^
Final viscosity/cp	3280.00 ± 6.00 ^a^	3292.00 ± 93.00 ^a^	3307.50 ± 6.50 ^a^	3261.50 ± 8.50 ^a^	3334.50 ± 8.50 ^a^
Setback value/cp	1621.50 ± 9.50 ^d^	1832.00 ± 52.00 ^c^	1960.00 ± 5.00 ^b^	2010.50 ± 12.50 ^ab^	2053.50 ± 8.50 ^a^
Peak time/min	5.60 ± 0.00 ^b^	5.73 ± 0.00 ^a^	5.70 ± 0.03 ^a^	5.73 ± 0.00 ^a^	5.73 ± 0.00 ^a^
Pasting temperature/°C	85.85 ± 0.00 ^c^	88.31 ± 0.38 ^b^	88.50 ± 0.20 ^b^	89.08 ± 0.07 ^b^	91.07 ± 0.31 ^a^

Different letters in the same row indicate a significant difference (*p* < 0.05).

**Table 3 foods-12-04509-t003:** Gelation properties of BRFs.

Sample	BRF100	BRF97	BRF94	BRF91	BRF89
Hardness/N	3.46 ± 0.01 ^a^	3.10 ± 0.05 ^a^	2.94 ± 0.01 ^ab^	3.03 ± 0.10 ^ab^	2.46 ± 0.35 ^b^
Adhesiveness /N.mm	1.67 ± 0.02 ^a^	1.24 ± 0.14 ^b^	1.09 ± 0.03 ^b^	1.05 ± 0.03 ^b^	0.76 ± 0.10 ^c^
Cohesiveness/g·s	0.52 ± 0.00 ^a^	0.53 ± 0.05 ^a^	0.60 ± 0.00 ^a^	0.58 ± 0.01 ^a^	0.59 ± 0.01 ^a^
Springiness/mm	7.65 ± 0.39 ^a^	6.41 ± 0.01 ^b^	6.13 ± 0.13 ^bc^	5.35 ± 0.15 ^c^	5.33 ± 0.22 ^c^
Gumminess/N	1.79 ± 0.31 ^a^	1.78 ± 0.06 ^a^	1.77 ± 0.23 ^a^	1.84 ± 0.09 ^a^	1.68 ± 0.28 ^a^
Chewiness/g·s	11.95 ± 0.45 ^a^	10.68 ± 0.05 ^b^	9.82 ± 0.10 ^bc^	9.09 ± 0.04 ^c^	8.90 ± 0.37 ^c^

Different letters in the same row indicate a significant difference (*p* < 0.05).

**Table 4 foods-12-04509-t004:** Cooking quality of BRNs.

Sample	Optimal Cooking Time/min	Water Absorption/%	Breaking Rate/%
BRN100	10.47 ± 0.08 ^a^	75.29 ± 1.65 ^a^	2.22 ± 1.57 ^a^
BRN97	10.23 ± 0.02 ^b^	66.03 ± 1.14 ^b^	7.78 ± 1.57 ^b^
BRN94	9.72 ± 0.21 ^c^	51.21 ± 0.44 ^c^	13.33 ± 2.72 ^c^
BRN91	9.17 ± 0.05 ^d^	48.21 ± 0.87 ^c^	17.78 ± 1.57 ^d^
BRN89	8.91 ± 0.07 ^e^	43.74 ± 0.67 ^d^	23.33 ± 2.72 ^e^
BRN1h	6.42 ± 0.04 ^a^	50.64 ± 1.64 ^a^	29.95 ± 1.64 ^a^
BRN3h	7.71 ± 0.13 ^b^	48.86 ± 0.92 ^a^	19.11 ± 1.14 ^b^
BRN5h	8.14 ± 0.14 ^c^	34.73 ± 0.68 ^b^	10.53 ± 0.43 ^c^
BRN7h	9.16 ± 0.16 ^c^	32.30 ± 1.40 ^b^	2.45 ± 0.87 ^d^
BRN9h	9.97 ± 0.14 ^d^	25.05 ± 2.30 ^c^	8.86 ± 0.67 ^c^

Different letters in the same row indicate a significant difference (*p* < 0.05).

**Table 5 foods-12-04509-t005:** Hardness, cohesiveness, and chewiness of BRNs.

Sample	Hardness/g	Cohesiveness/g·s	Chewiness/g·s
BRN100	3001.44 ± 65.91 ^a^	0.70 ± 0.00 ^a^	1976.87 ± 41.34 ^a^
BRN97	2665.99 ± 61.74 ^a^	0.69 ± 0.00 ^ab^	1726.56 ± 37.45 ^a^
BRN94	2257.61 ± 49.51 ^b^	0.68 ± 0.00 ^ab^	1437.74 ± 89.00 ^b^
BRN91	1786.93 ± 9.20 ^c^	0.68 ± 0.00 ^bc^	1122.11 ± 7.34 ^c^
BRN89	1300.62 ± 54.15 ^d^	0.66 ± 0.01 ^c^	765.78 ± 134.28 ^d^
BRN1h	1022.05 ± 20.15 ^c^	0.68 ± 0.00 ^a^	626.05 ± 12.23 ^d^
BRN3h	1048.41 ± 1.51 ^c^	0.70 ± 0.01 ^a^	677.98 ± 13.74 ^c^
BRN5h	1181.17 ± 11.55 ^b^	0.69 ± 0.01 ^a^	739.18 ± 10.85 ^b^
BRN7h	1188.98 ± 13.75 ^b^	0.69 ± 0.00 ^a^	749.48 ± 2.23 ^b^
BRN9h	1311.40 ± 41.33 ^a^	0.69 ± 0.01 ^a^	824.64 ± 15.81 ^a^

Different letters in the same row indicate a significant difference (*p* < 0.05).

**Table 6 foods-12-04509-t006:** The color measurement of BRNs.

Sample	L	a	b	W
BRN100	89.96 ± 0.74 ^a^	11.63 ± 0.36 ^ab^	6.82 ± 0.27 ^a^	83.19 ± 0.74 ^a^
BRN97	88.38 ± 0.18 ^a^	11.40 ± 0.28 ^ab^	8.40 ± 0.02 ^b^	81.68 ± 0.18 ^a^
BRN94	85.08 ± 0.21 ^b^	11.88 ± 0.29 ^a^	13.27 ± 0.28 ^c^	76.76 ± 0.21 ^b^
BRN91	80.26 ± 1.63 ^c^	11.54 ± 0.08 ^ab^	14.65 ± 0.79 ^d^	72.84 ± 1.63 ^c^
BRN89	79.51 ± 0.48 ^c^	11.16 ± 0.15 ^b^	17.51 ± 0.42 ^e^	70.82 ± 0.48 ^c^
BRN1h	88.35 ± 0.41 ^a^	5.02 ± 0.57 ^a^	12.24 ± 0.32 ^a^	82.36 ± 0.55 ^a^
BRN3h	87.15 ± 0.04 ^b^	4.58 ± 0.35 ^b^	12.79 ± 0.30 ^a^	81.28 ± 0.28 ^b^
BRN5h	85.49 ± 0.40 ^b^	4.15 ± 0.15 ^b^	13.54 ± 0.17 ^b^	79.73 ± 0.35 ^c^
BRN7h	84.50 ± 0.45 ^d^	4.16 ± 0.59 ^b^	14.1 ± 0.19 ^c^	78.47 ± 0.13 ^d^
BRN9h	82.67 ± 0.42 ^e^	2.31 ± 0.05 ^c^	14.18 ± 0.10 ^d^	77.49 ± 0.38 ^e^

Different letters in the same row indicate a significant difference (*p* < 0.05).

**Table 7 foods-12-04509-t007:** Sensory evaluation of BRNs.

Sample	Firmness	Chewiness	Elasticity	Slipperiness	Overall Acceptability	Total
BRN100	19.33	18.27	18.00	18.40	19.73	93.73
BRN97	17.47	17.47	17.60	16.27	17.73	86.53
BRN94	16.40	14.27	15.87	16.00	16.93	79.47
BRN91	14.00	12.27	13.73	14.80	16.13	70.93
BRN89	12.53	11.47	11.60	12.53	13.73	61.87
BRN1h	11.97	14.00	12.47	10.00	11.00	59.43
BRN3h	12.43	12.80	13.03	12.90	12.00	63.17
BRN5h	14.57	13.87	13.97	13.83	13.33	69.57
BRN7h	18.33	13.00	15.90	16.40	16.33	79.97
BRN9h	16.23	13.83	14.43	15.40	14.97	74.87

## Data Availability

All related data and methods are presented in this paper. Additional inquiries should be addressed to the corresponding author.

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
