# Peer review of "Effect of Rice Bran and Retrograded Time on the Qualities of Brown Rice Noodles: Edible Quality, Microstructure, and Moisture Migration"

_foods, 2023, doi:10.3390/foods12244509_

Round 1

Reviewer 1 Report

Comments and Suggestions for Authors

The topic of using bran as a food ingredient is valuable. The study was well designed and the manuscript is well organised. Yet, some tables should be improved. More detail comments are given below.

The abstract should be improved. The authors should provide some info on the methodology used and more results (numbers).

Line 48 explain the abbreviation GI

L 53: I do not understand the meaning of 'squeezing' in this context.

L 80 white rice flour

Line 120-121: Micro (typing error); what was the test speed?

Lines 126 and 131: what where the criteria for similar quality?

Line 127: what was the temperature of water?

Table 2 please provide the measuring unit for chewiness; the springiness is usually in range between 0 and 1 - please explain.

Table 5: I suggest to use SI unit N instead of gram

Conclusion: try to avoid the use of abbreviations.

Comments on the Quality of English Language

There is a need for English language editing, since many sentences are incomplete or have awkward structure (e.g. lines 15-16; 50-52). 

Reviewer 2 Report

Comments and Suggestions for Authors

The study explores the physicochemical properties and culinary quality of brown rice noodles (BRNs) by incorporating varying amounts of rice bran and adjusting the retrograded time, aiming to enhance the nutritional and textural characteristics of these whole-grain products. The findings reveal several significant alterations in the properties of brown rice flours (BRFs) due to the addition of rice bran. These changes include modifications in pasting properties, gel properties, and texture of BRFs.

The introduction of rice bran into the BRNs is associated with notable effects on cooking time, water absorption, and various textural attributes. Specifically, the presence of rice bran reduces the optimal cooking time and water absorption while increasing the breaking rate. Furthermore, the addition of rice bran contributes to a decrease in hardness, chewiness, and crystallinity of the noodles. The altered water dynamics within the BRNs result in a looser structure, with decreased binding water and increased free water content.

The study also underscores the impact of retrograded time on the quality of BRNs. Notably, a retrograded time of 7 hours yields cooked BRNs with a lower breaking rate, optimal hardness, cohesiveness, and chewiness. The resulting structure is more compact, with higher internal binding water content and lower free water content.

These findings provide valuable insights for the development of nutritionally enriched and high-quality rice flour products. The study offers a theoretical foundation for the industrial production of BRNs, suggesting potential avenues for creating healthier and appealing whole-grain noodle products.

Key properties, including cooking characteristics and texture of the resulting BRNs, were extensively examined. The microstructure and water migration of BRNs were also subjects of investigation. The overarching goal of the study is to establish a theoretical foundation for the incorporation of BRNs into commercial production, ultimately promoting the consumption of brown rice over white rice.

However, some comments in order to improve the manuscript:

While the study presents a systematic exploration of the effects of rice bran content and retrograded time on BRNs, the critical review would benefit from more detailed information on specific findings and their significance.

 A clearer presentation of the observed physicochemical changes and their implications for the commercial production and consumption of brown rice noodles would enhance the depth of the manuscript.

In the conclusion of this study, it is evident that the addition of rice bran significantly influences the physicochemical properties of rice flour and, consequently, brown rice noodles (BRNs). The observed effects encompass various aspects, including the pasting viscosity of starch, the formation of starch gel, and the hardness, elasticity, and adhesiveness of the starch gel. As the proportion of rice bran increases, several notable changes are observed in the properties of BRNs. However, the lack of a correct sensory analysis (flavour, odour, texture, colour..etc), in order to know if this product (noodles) would have a good acceptability between consumers, is a problem and should be included into the study.

IN addition, the lack of colour measurement and industrial application makes difficult to understand the study.

Reviewer 3 Report

Comments and Suggestions for Authors

This is a manuscript looking at rice bran addition to white rice to improve the edible quality, micro-structure and moisture migration of brown noodles. It is written in good english, but in some places need special attention to deliver the correct meaning of the sentences. Also typo errors and grammatical errors should be checked.

Noodles were prepared by mixing quantities of bran with white rice with different proportions. I cannot see anywhere optimum ratio of these two was discussed and given. Also edible quality of the noodles should be checked with a sensory panel evaluation rather than expressing with instrumental readings.

Moisture migration should be expressed as a quantitative equation so it is easy to translate rations into output quantities.

Also involvement of several variables need to incorporate surface response method evaluation to determine the optimum conditions. If possible Baysean method could be adopted.

Also quantify micro-structure  using some method like image processing.

+

Comments on the Quality of English Language

Minor corrections of typo and grammar needed

Round 2

Reviewer 2 Report

Comments and Suggestions for Authors

Accept

Author Response

Thank you very much for your review again.  

Reviewer 3 Report

Comments and Suggestions for Authors

Thanks for answering the questions and comments and doing necessary actions appropriate for the manuscript. Appreciated.

Author Response

(The authors gave the same response as above.)
